# Natural Vegetation Area Design in an Arid Region Based on Water Resource Carrying Capacity—Taking Minqin County as an Example

**Hengjia Zhang** [1,*,†], **Jiandong Yu** [1,†], **Tianliang Jiang** [2], **Shouchao Yu** [1], **Chenli Zhou** [1], **Fuqiang Li** [2] **and Xietian Chen** [1]

1   College of Agronomy and Agricultural Engineering, Liaocheng University, Liaocheng 252059, China; jiandongyu99@163.com (J.Y.); ysc@lcu.edu.cn (S.Y.); zhouchenli2021@126.com (C.Z.); gsaucxt@163.com (X.C.)
2   College of Water Conservancy and Hydropower Engineering, Gansu Agricultural University, Lanzhou 730070, China; jiangtl@iwhr.com (T.J.); lifq@gsau.edu.cn (F.L.)
*   Correspondence: zhanghengjia@lcu.edu.cn
†   These authors contributed equally to this work.

**Abstract:** Water resource management and natural vegetation distribution status are important for the sustainable development of agricultural, ecological and socio-economic systems in arid areas, and the carrying area of vegetation is limited by the established water resources. This study proposed the concept of natural vegetation area design in arid areas based on water resource design carrying capacity and took Minqin County belonging to a typical inland arid area as an example and socio-economic water consumption, ecological water consumption and water resource utilization efficiency in the study area as the main selected factors as well as reference indicators. By calculating the water ecological footprint, water resource carrying and water resource ecological pressure index of the main natural vegetation design area in Minqin County in 2017, we analyzed and evaluated its water resource carrying status and predicted the natural vegetation areas in 2025. The results showed that there was a large gap (the increase was 13.25–9.29%) between the actual area of various types of natural vegetation in 2017 and that in 2025. The water resource utilization was more effective in various types of natural vegetation in Minqin County under the stable development model, and the water ecological deficit was approaching 0, but the utilization of forest trees other than shrubs and herbaceous plants was in an insecure state, and the water ecological deficit was less than 0. The water resource allocation schemes under the restoration model and the optimization model were safer ecological development models for natural vegetation in Minqin County, which was in a state of water-ecological surplus, and the water-ecological surplus value was greater than 0. Thus, it could be seen that coordinating the water resource supply among various types of natural vegetation in an arid area was a preferred strategy to ensure the sustainable development of regional ecology as well as an effective countermeasure to improve the water use efficiency of natural vegetation in the county to a certain extent. This study aimed to evaluate the carrying capacity of a natural vegetation area and the water pressure index under certain water supply conditions, which could provide a reasonable theoretical reference for water resource management in arid areas.

**Keywords:** water carrying capacity; natural vegetation; water resource management; arid region; water resource ecological pressure index

## 1. Introduction

Water is an important resource for the sustainable development of arid area vegetation as well as a key natural resource for regional ecological security and socio-economic development [1]. Due to the continuous exploitation of water resources, the excessive occupation of the ecological water and the frequent occurrence of extreme weather caused by climate change, the ecological environment in arid areas is increasingly fragile, which

seriously restricts sustainable development at both the national and regional levels [2]. As human society's desire for continuous development grows stronger, the contradiction between economic development and the balanced utilization of water resources is becoming more and more prominent [3,4]. Balancing the issues of water resource development and conservation under climate change and ecologically sustainable development conditions is particularly important for maintaining a balance between the water environment and the economy as well as a sustainable supply of water resources in arid regions [5]. The water resource carrying capacity, ecological sustainability and response measures caused by global climate change have been widely concerned and studied by many institutions and scholars [6,7]. The United Nations Intergovernmental Panel on Climate Change (IPCC) proposed in "*The report Climate Change 2022: Impacts, Adaptation and Vulnerability*" that "healthy ecosystems are more resilient to climate change and will provide adequate food, water and other essential basic services". For the arid and water-scarce regions of northwest China's oases, natural vegetation conservation and design become important pathways for regional water conservation, food security, ecological environmental security and maximum water resource carrying capacity.

Water resource carrying capacity (WRCC) is an exploration of the response mechanism between human activities and water resources and is a rational assessment method for the threshold of the socio-economic scale that water resources can carry, accompanied by the rational allocation of limited water resources [8]. It was used in the 1980s to study the relationship between economic wealth and land productivity [9]. At present, the concept of WRCC has been widely used in water supply research between industrial and economic development in arid water-scarce areas, as well as in the analysis of water demand balance for industrial [10], agricultural [11], ecological [12] and domestic [13] purposes, etc. The research mostly focuses on the improvement and application of water resource carrying capacity research methods and evaluation systems [14] and the prediction of future WRCC [15,16].

The current WRCC is relatively easy to calculate when the total water resources and socio-economic development patterns are known. However, the total amount of water resources is influenced by a number of factors and changes over time, and the socio-economic development patterns under different conditions must be estimated at different times [17]. Therefore, it is difficult for the current WRCC calculation and evaluation system to make accurate predictions about the future. To reduce the uncertainty in future water resource carrying capacity calculations, the concept of water resource design carrying capacity (WRDCC) was proposed [18].

WRDCC is the maximum population that can be supported by water resources under the designed socio-economic development pattern and water supply conditions [18]. Currently, the design of vegetation planning areas in China is mainly based on qualitative analysis in terms of weather, topography, land use types, etc., and elaborates on the impacts on the environment, climate and vegetation in terms of population interventions, which lacks quantitative scientific calculations [19,20]. This may result in unscientific vegetation area planning due to subjective evaluations [21,22]. Therefore, adopting objective and relatively accurate calculation methods to rationalize the allocation of limited water resources and at the same time designing appropriate development modes and thus restoring the vegetation and ecology of arid zones, so as to reduce the uncertainty in the process of calculating the water resource carrying capacity in the traditional sense, is the key to solve this problem [18,23]. In this study, the concept and evaluation system of the natural vegetation water resource design carrying area (NVDA) is proposed based on WRDCC, taking a typical arid area of Minqin County in Gansu Province, China, as a case study to reduce the uncertainty of future water resources and provide a suitable NVDA calculation method. Therefore, the objective of the present study was to control the water consumption in human production within the scope of water ecology and the water cycle after analyzing the exploitation and utilization of water resources in Minqin County and similar areas,

which would provide a theoretical basis for the establishment of social and economic systems and the construction of an ecological environment in arid areas.

## 2. Materials and Methods

### 2.1. Study Area

Minqin County is located in the lower reaches of Shiyang River in Gansu Province, Northwest China, between $101°49'\sim104°12'$ E and $38°03'\sim39°28'$ N (Figure 1). It is surrounded by the Badain Jaran Desert and the Tengger Desert in the west, north and east. The total area is $15.9 \times 10^3$ km$^2$ (desertification area accounts for 90.34%). The region has a typical temperate continental climate, dry with little rain, strong evaporation, large daily changes in temperature and the county's annual average temperature of 7.8 °C. The average annual precipitation is 115 mm, mainly concentrated in July–September, with a potential annual evaporation of 2644 mm. The common flora consists of arbors, shrubs and herbs. Arbors include Populus euphratica. Shrubs include Nitraria tangutorum, Sacsaoul and Calligonum. Herbs include Suaedaforsk, Bassia dasyphylla and Echinops sphaerocephalus. The maximum groundwater depth of 4.5 m in the study area is based on the average value when the groundwater was undisturbed around 1980 [24]. In the past 20 years, the groundwater level in some areas of the county has decreased at a rate of 0.6 $m/a$, and groundwater mineralization has increased at a rate of 0.1 $g/(L-a)$ [25].

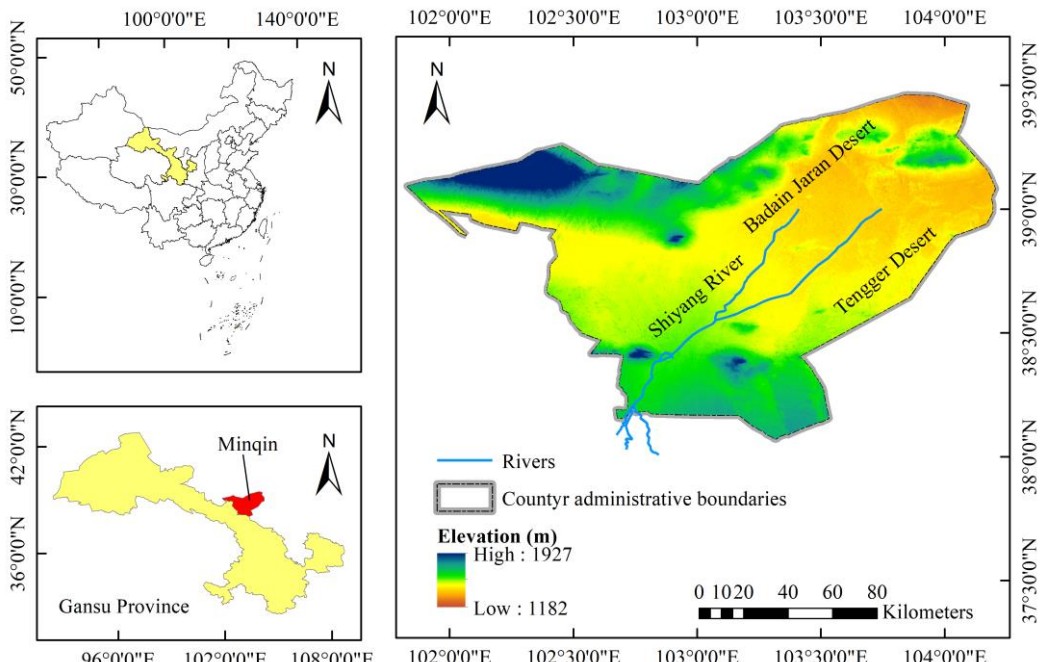

**Figure 1.** Schematic diagram of the study area.

### 2.2. Data Collection

According to the "*Wuwei Statistical Yearbook*" and "*The Seventh National Census Bulletin of Minqin County*", the main data required for the design of the natural vegetation area for the WRCC of the study area were collected. In 2017, the county's resident population was $24.18 \times 10^4$, of which the urban population was $8.3 \times 10^4$, and the rural population was $15.88 \times 10^4$. The total available water resources in the county were $3.58 \times 10^8$ m$^3$. The gross regional product was USD $10.53 \times 10^8$. The added value of the primary industry was USD $3.73 \times 10^8$, up 6.18%, the added value of the secondary industry was USD $2.73 \times 10^8$, down 12.90%, and the added value of the tertiary industry was USD $4.06 \times 10^8$, up 4.50%. According to the "*Notice of the People's Government of Minqin County on the Issuance of the Fourteenth Five-Year Plan of National Economic and Social Development of Minqin County and*

*the Outline of Visionary Goals for 2035"*, by 2025, the total available water resources and the available water supply of the county will be $4.13 \times 10^8$ m$^3$. According to *"Minqin County Rural Domestic Sewage Treatment Special Plan (2022–2030)"* and Equation (1), the county's population in 2025 is predicted to be

$$P = P_o(1 + x)^n \tag{1}$$

where $P_o$—total population of the county in 2020, and $P_o = 17.74 \times 10^4$ people; $P$—total population of the county in 2025; $x$—the average annual population growth rate of Minqin County is $-2.97\%$; and $n$—number of measurement years, $n = 5$.

"*Minqin County 'fourteen five' water development plan*" shows that by 2025, the county's water supply guarantee rate will be 90%, and the county's population will be $15.26 \times 10^4$ people. The average annual growth of gross regional product will reach 6.5%, the added value of the primary industry will be USD $7.50 \times 10^8$, the added value of the secondary industry will be USD $1.75 \times 10^8$, and the added value of the tertiary industry will be USD $6.35 \times 10^8$.

### 2.3. Data Processing

The specific meaning of NVDA in this study is the maximum carrying area of water resources for natural vegetation under the design socio-economic development model. The meaning of " water supply design" here is similar to that of "design flood" but has different characteristics and meanings from that of "general flood". The design flood is the maximum flood in flood control and flood forecasting and is expected to require reinforcement. Similarly, the NVDA and design characteristics are fixed when determining the design quantity of water resources and socio-economic development patterns.

#### 2.3.1. Water Supply Design for NVDA Solutions

The design water supply is a key step in calculating NVDA and is the basis for establishing the NVDA scheme. In this study, the design water supply ($W_P$) is the possible amount of water resources in a certain period of time in the future. The probability of $W_P$ is $P$. That is, when the water supply is greater than or equal to $W_P$, the WRCC guarantee rate of the natural vegetation areas is $P$, and the risk of not meeting the design standard is $1 - P$. It is particularly important to assign a precise value to $P$, due to the influence of socio-economic diversity and local policies. The study found [18,26] that $P$ can be assigned a value (Table 1). The water supply guarantee rate is generally greater than 95% for residential and 90% for industrial areas. The lower guarantee rates in rural areas are due to natural and economic constraints.

**Table 1.** Design guarantee rate, P(%).

| Region Type | Small-Scale Region | Mid-Scale Region | Large-Scale Region |
|---|---|---|---|
| Water-rich area | 85% | 90% | 95% |
| Water-scarce area | 80% | 85% | 90% |

#### 2.3.2. Development Model Design for NVDA Solution

In this study, the design development approach reflects the characteristics of the relationship between the ecological condition of natural vegetation and water resource use, which in turn is closely related to economic policies, development planning and basic state policies. This includes the priority of regional development strategies on water demand for ecological and environmental protection, as well as the influence of policies, regulations and management systems on the water resource carrying capacity. We proposed three types of development models based on the level of ecological development in the arid areas:

(1) Stabilization model. This model maintains the existing vegetation area and abundance in a good state and ensures that the ecological environment will not deteriorate further.

(2) Restoration model. This model restores the vegetation in the arid areas to a certain extent and promotes the benign evolution of the regional ecology.

(3) Optimization mode. The type and quantity of regional vegetation can reach the maximum carrying capacity. At the same time, the ecological and environmental functions can give full play to their potential.

The carrying capacity varies with the development pattern. Therefore, determining the development pattern of a certain region is one of the important steps in calculating NVDA.

### 2.3.3. Evaluation Indicators in Development Model Evaluation

To accurately evaluate the design and development approach, the selection of indicators should be based on the following principles: (1) significant impact on ecological water use, (2) clear conceptual and physical meaning and (3) easy to compare and calculate. Based on previous studies, the indicators selected for evaluation are mainly socio-economic systems, ecosystems and water use (Figure 2).

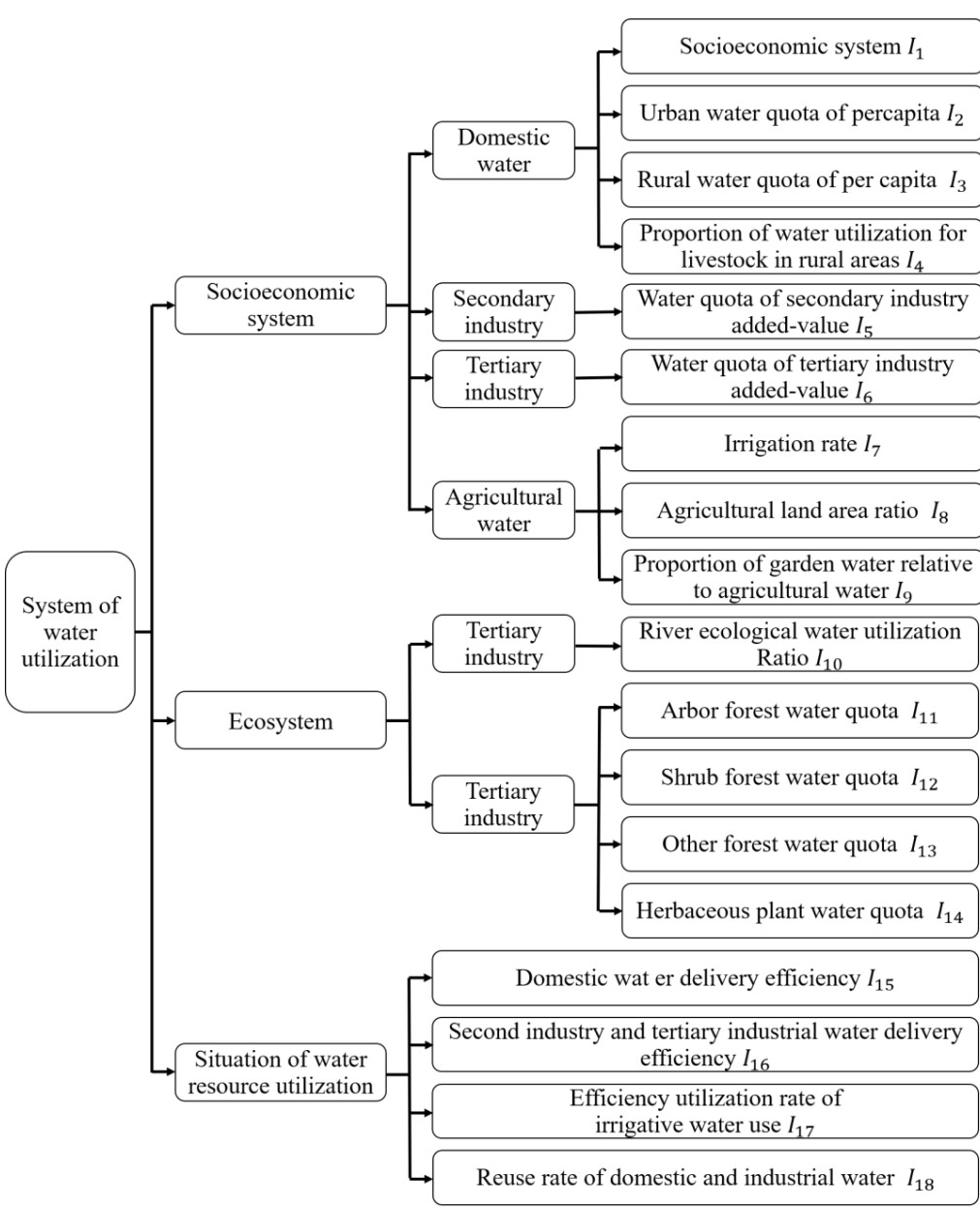

**Figure 2.** Evaluation index of water use system.

### 2.3.4. Vegetation Water Demand Quota Calculation

Soil moisture is one of the main limiting factors for plant growth and development, and it is determined by the intensity of evaporation. If soil moisture evaporates steadily, the soil water content and evaporation intensity of surface moisture will remain stable [27]. Different vegetation types have different optimal groundwater burial depths, so water demand quotas can be calculated by the product of groundwater evaporation intensity and vegetation coefficient [28].

$$W = E \times K \tag{2}$$

where $W$—vegetation water demand quotas; $E$—groundwater evaporation intensity; and $K$—vegetation factor.

### 2.3.5. Groundwater Evaporation Intensity Calculation

Two different methods can be used for the calculation of groundwater evaporation intensity ($E$). The first method includes Penman–Monteith model and Hargreaves model, etc., which are developed based on meteorological factors, such as rainfall, temperature, solar radiation, etc. The second method includes a model based on the relationship between groundwater evaporation and depth. The latter is more suitable for arid areas where the test conditions are poor and the parameters are difficult to obtain. These methods can generally be further distinguished into the following three subcategories: (1) Groundwater $E$ has only a single correlation with groundwater depth of burial, such as through Ye's index equation [29]. This type of model has some limitations because it ignores the law of $E$ related to evaporation. (2) The relationship between $E$ and evaporation, such as Shen's equation [30]. (3) The relationship between $E$ and soil moisture and evaporation from the soil surface, as in the Lei Zhi-Dong equation [31]. Studies in arid areas show that the latter two model subclasses fit the measured data better when the groundwater burial depth is greater than 1 m [32].

Due to the different conditions set by various models, the calculation of $E$ with a single equation appears to be inaccurate. Using MATLAB software, the extreme value rejection method [33] and the golden section search method [34] are popular and widely optimized methods. Therefore, we use these methods to select the optimal results from the seven classical equations. The specific programming consists of the following three steps:

(1) Assuming that the extreme value of groundwater burial depth ($h_0$) in the study area is $H$ (the extreme value of groundwater burial depth in Minqin County is 4.5 m), the interval and step size are [1, $H$] and 0.01, respectively. Then, the values of $n$ in the interval are substituted into the equations described in Table 2, and the results can be formed into the matrix ($E$):

$$E = \begin{bmatrix} E_{1,1} & E_{1,2} & \cdots & E_{1,7} \\ E_{2,1} & E_{2,2} & \cdots & E_{2,7} \\ \cdots & \cdots & \cdots & \cdots \\ E_{n,1} & E_{n,2} & \cdots & E_{n,7} \end{bmatrix} \tag{3}$$

where $n$—total number of equation results, and $n = (H - 1)/0.01$.

(2) Removing the maximum and minimum values in each row of matrix $E$ to obtain the new matrix ($E'$).

$$E' = \begin{bmatrix} E'_{1,1} & E'_{1,2} & \cdots & E'_{1,5} \\ E'_{2,1} & E'_{2,2} & \cdots & E'_{2,5} \\ \cdots & \cdots & \cdots & \cdots \\ E'_{n,1} & E'_{n,2} & \cdots & E'_{n,5} \end{bmatrix} \tag{4}$$

(3) The optimal value ($E_{OPi}$) can be calculated by a simple mathematical equation called the golden mean, which has the following expression.

$$E_{OPi} = \max(E'_i) \times 0.618 + min(E'_i) \times 0.382 \quad i = 1, 2, 3, \cdots, n. \tag{5}$$

**Table 2.** Potential evapotranspiration equations.

| Formula Number | Formula Name | Formula Form | References |
|:---:|:---:|:---:|:---:|
| 1 | Shen | $E = c\mu E_0^a/(1+h)^b$ | [30] |
| 2 | Ye | $E = E_0/e^{-ah}$ | [29] |
| 3 | Lei | $E = ah^{-b}/\left(1 - e^{-0.85E_0/ah^{-b}}\right)$ | [31] |
| 4 | Mao | $E = E_0(1 - (h - \Delta h)h_0)^a$ | |
| 5 | Zhang | $E = E_0a(h_0 + N)^b$ | |
| 6 | Anti-log | $E = cE_0/ae^{bh}$ | [35] |
| 7 | Shu | $E = aE_0^{(1+b)}(1 - h/h_0)^c$ | |
| 8 | Averyanov | $E = E_0a(1 - h/h_0)^b$ | |

Note: $a$, $b$, $c$ is an empirical parameter. $E_0$ is the evaporation from the water surface of 601 evaporation dishes, mm/d; $h$ is the buried depth of groundwater, m.

2.3.6. Vegetation coefficient calculation

The vegetation coefficient ($K$) reflects the influence of vegetation on groundwater evaporation in a certain area and is obtained by dividing the groundwater evaporation in the vegetated area by that in the non-vegetated area. Finding table is the conventional method to determine $K$. Since this method is limited to the values contained in the table, it is obviously inaccurate for the calculation of $K$. Therefore, in this study, by regressing the experimental data (shown in Table 3) and implementing the second-order Gaussian model in MATLAB, the following equation (6) for calculating $K$ is obtained:

$$K = 1353.5exp\left(-\left(\frac{h + 16.59}{6.79}\right)^2\right) + 1.07exp\left(-\left(\frac{h - 3.13}{1.92}\right)^2\right) \tag{6}$$

**Table 3.** Relationship between groundwater depth of burial and vegetation coefficient.

| Groundwater depth (m) | 1.0 | 1.5 | 2.0 | 2.5 | 3.0 | 3.5 | 4.0 |
|:---|:---:|:---:|:---:|:---:|:---:|:---:|:---:|
| **Vegetation coefficient** | 1.98 | 1.63 | 1.56 | 1.45 | 1.38 | 1.29 | 1.00 |

Note: The values of vegetation coefficients used are taken from the results of experiments in Yumen Town and the Hexi Corridor [36].

2.3.7. NVDA Calculation

According to the water balance principle, the water balance equation is established, and the NVDA is calculated. The water demand quotas for all vegetation types can be obtained by using the above method. Assuming that the areas of the four main vegetation types that water resources can support are $X_1$, $X_2$, $X_3$ and $X_4$, we solve Equations (7)–(24).

Domestic water $Q_1$ ($10^4$ m$^3$):

$$Q_1 = Q_{11} + Q_{12} + Q_{13} \tag{7}$$

Urban domestic water $Q_{11}$ ($10^4$ m$^3$):

$$Q_{11} = 365 \times TP \times I_1 \times I_2/I_{15} \tag{8}$$

Rural domestic water $Q_{12}$ ($10^4$ m$^3$):

$$Q_{12} = 365 \times TP \times (1 - I_1) \times I_3/I_{15} \tag{9}$$

Livestock water $Q_{13}$ ($10^4$ m$^3$):

$$Q_{13} = Q_{12} \times I_4 \tag{10}$$

Secondary industry water $Q_2$ ($10^4$ m$^3$):

$$Q_2 = SIA \times I_5 / I_{16} \tag{11}$$

Tertiary industry water $Q_3$ ($10^4$ m$^3$):

$$Q_3 = TIA \times I_6 / I_{16} \tag{12}$$

Agriculture water $Q_4$ ($10^4$ m$^3$):

$$Q_4 = Q_{41} + Q_{42} \tag{13}$$

Irrigation water $Q_{41}$ ($10^4$ m$^3$):

$$Q_{41} = A \times I_7 \times I_8 / I_{17} \tag{14}$$

Garden water $Q_{42}$ ($10^4$ m$^3$):
$$Q_{42} = Q_{41} \times I_9 \tag{15}$$

Socio-economic water $Q_S$ ($10^4$ m$^3$):

$$Q_S = (Q_1 + Q_2 + Q_3 + Q_4) \times (1 - I_{18}) \tag{16}$$

Natural vegetation ecological water $Q_E$ ($10^4$ m$^3$):

$$Q_E = Q_{E1} + Q_{E2} + Q_{E3} + Q_{E4} \tag{17}$$

Arbor forest ecological water $Q_{E1}$ ($10^4$ m$^3$):

$$Q_{E1} = I_{11} \times X_1 \tag{18}$$

Shrub forest ecological water $Q_{E2}$ ($10^4$ m$^3$):

$$Q_{E2} = I_{12} \times X_2 \tag{19}$$

Other forest ecological water $Q_{E3}$ ($10^4$ m$^3$):

$$Q_{E3} = I_{13} \times X_3 \tag{20}$$

Herbaceous plant ecological water $Q_{E4}$ ($10^4$ m$^3$):

$$Q_{E4} = I_{14} \times X_4 \tag{21}$$

River ecological water $Q_R$ ($10^4$ m$^3$):

$$Q_R = Q_E \times I_{10} / (1 - I_{10}) \tag{22}$$

Total ecological water $Q_T$ ($10^4$ m$^3$):

$$Q_T = Q_R + Q_E \tag{23}$$

Total water consumption $Q$ ($10^4$ m$^3$):

$$Q = Q_S + Q_T \tag{24}$$

where $TP$—the total population in the study area; $SIA$—the secondary industry added values; $TIA$—tertiary industry added values; $Q$—the design of water supply (refer to Table 1), and $Q = W_{80\%} = 4.13 \times 10^8$ m$^3$; $X_1$—area of arbor forest in the Minqin County; $X_2$—area of shrub forest in the Minqin County; $X_3$—area of other forests in the Minqin

County; $X_4$—area of herbaceous plants in the Minqin County; and $A$—irrigated farmland in Minqin County.

2.3.8. NVDA Evaluation

The water ecological footprint (WEF) and water resource carrying capacity (WRCC) were used to evaluate the status of non-point source pollution. By definition, the WEF is the biologically productive land area necessary to sustain water consumption and absorb water pollution for a given population and economic scale [37]. We extended the water footprint to natural vegetation in arid areas, reflecting the proportional use of water resources by natural vegetation, by applying Equation (25) to the WEF.

$$EF_W = \psi_W \times \left( \frac{C_W}{P_W} \right) \tag{25}$$

where $EF_W$—ecological footprint of water resources in natural vegetation; $\psi_W$—the water balance factor, taking the value of 5.19; $C_W$—vegetation water consumption, m$^3$/hm$^2$; and $P_W$—the average production capacity of water resources, taking the value of $3.14 \times 10^3$ m$^3$/hm$^2$.

The WRCC Equation (26) based on WEF can be expressed as follows:

$$EC_W = \pi \times \phi_W \times \psi_W \times \left( \frac{Q_W}{P_W} \right) \tag{26}$$

where $\pi$—water resource development utilization rate, which takes the value of 40%; $EC_W$—water carrying capacity of natural vegetation, hm$^2$; $\phi_W$—the water yield factor, which takes the value of 0.22; $P_W$—the average production capacity of water resources, taking the value of $3.14 \times 10^3$ m$^3$/hm$^2$; and $Q_W$—design water demand of natural vegetation, m$^3$/hm$^2$.

As can be seen from the previous section, WEF reflects the consumption of water resources by human activities, while WRCC reflects the amount of water provided by humans. The water ecological deficit (or surplus) is obtained from the difference between WEF and WRCC, as shown in Equation (27).

$$ED_W(ES_W) = EC_W - EF_W \tag{27}$$

where $ED_W$—water ecology deficit; and $ES_W$—water ecology surplus. When the difference is less than 0, $ED_W$ indicates that the water resources have been over-utilized and exceeded the bearable threshold. Conversely, $ES_W$ indicates the extent to which water resources are underutilized.

Here, the water resource ecological pressure index (WREPI) is defined as the ratio of WEF to WRCC. Thus, WREPI can be quantified by using Equation (28).

$$EPI_W = EF_W / EC_W \tag{28}$$

where $EPI_W$—water resource ecological pressure index. To assess NVDA, we classified WREPI into three basic types: ecologically safe, ecologically alarming and ecologically unsafe. We then divided each type into subtypes (Table 4).

**Table 4.** WREPI grades.

| Type | Subtype | WREPI |
|---|---|---|
| | I | <0.25 |
| Ecological safety | II | 0.25–0.50 |
| | III | 0.50–0.75 |
| Ecological alarm | I | 0.75–1.00 |
| | II | 1.00–1.25 |
| | I | 1.25–1.50 |
| Ecological insecurity | II | 1.50–1.75 |
| | III | >1.75 |

## 3. Results and Analysis

### 3.1. Preferred Method for Calculating Groundwater Evaporation Intensity

In the process of calculating the groundwater evaporation intensity, we selected three classical equations for comparison. The correlation analysis and the difference test results (Table 5) between the fitted results of the three equations and the measured points showed that (1) the correlation coefficient ($R$) between the optimized results and the measured results was the largest, and the correlation coefficient ($R$) between Ray's results and the measured results was the smallest.

**Table 5.** Correlation analysis and difference test between the results of classical equation fitting and Zhang's results.

| Equation | Phreatic Water Evaporation at Different Depths (m) | | | | | | | | $R$ | $R_{0.05}$ | $P$ |
| | 1 | 1.5 | 2 | 2.5 | 3 | 3.5 | 4 | 4.25 | | | |
|---|---|---|---|---|---|---|---|---|---|---|---|
| Shen's equation | 1042.89 | 617.84 | 325.75 | 264.10 | 153.24 | 115.88 | 87.66 | 85.38 | 0.92 | 0.71 | 0.91 |
| Ye's equation | 1056.18 | 669.79 | 345.80 | 268.16 | 125.05 | 79.13 | 47.58 | 45.23 | 0.92 | 0.71 | 0.90 |
| Le's equation | 1056.88 | 624.00 | 319.03 | 258.77 | 120.76 | 104.08 | 92.76 | 83.25 | 0.91 | 0.71 | 0.91 |
| Optimization results | 1051.11 | 652.30 | 338.14 | 266.61 | 142.48 | 101.84 | 72.35 | 70.04 | 0.93 | 0.71 | 0.93 |
| Zhang's results | 993.10 | 706.86 | 360.60 | 274.55 | 163.59 | 150.27 | 94.34 | 83.13 | 1.00 | 0.71 | 1.00 |

(2) The $p$-value of the optimized results is the largest, and the $p$-value of Ye's equation is the smallest. The fitting results and optimized values of the three classical equations were obtained, and the optimized results were reasonable (Figure 3).

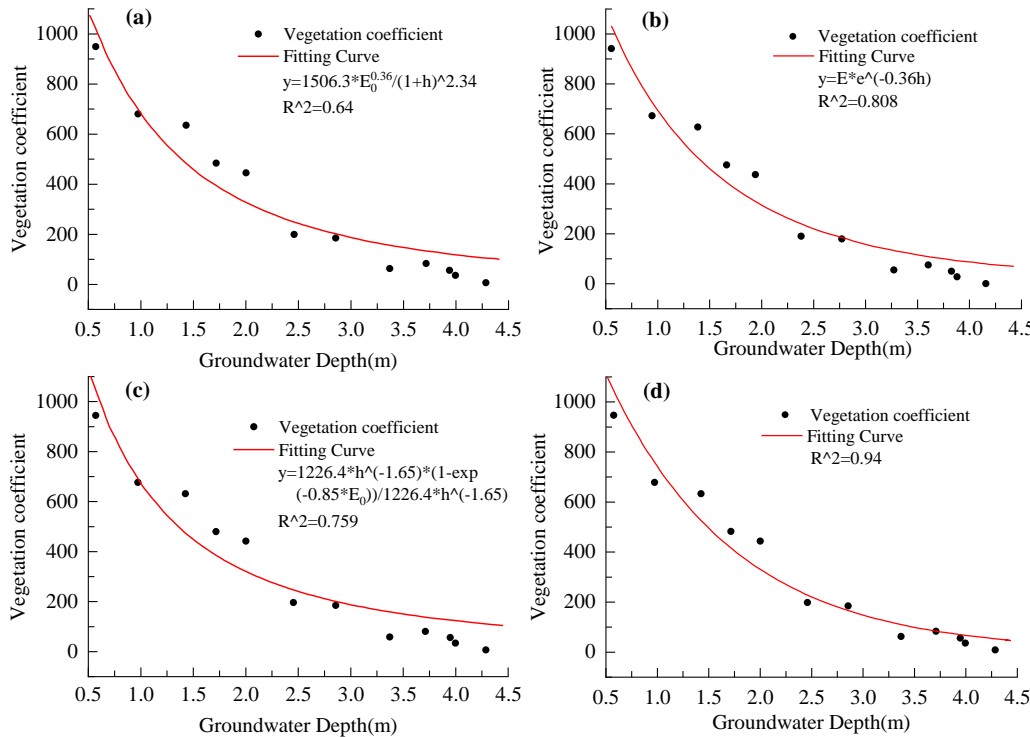

**Figure 3.** Relationship between groundwater evaporation and groundwater depth of burial. (**a**) Fitting results of Shen's equation. (**b**) Fitting results of Ye's equation. (**c**) Fitting results of Lei's equation. (**d**) Optimal results.

In this study, the calculation of groundwater $E$ gives optimized results without specific formulas from numerous equations. The groundwater burial depth evaporation curve is a fold line composed of a large number of optimized points, and when the burial depth is shallow, Shen's formula will have a greater impact on the $E$ value. When the burial depth is

greater than 3.5 m, the groundwater evaporation decline rate is determined by Ye's and Lei's formula, which is more suitable for calculating this burial depth condition. Therefore, the optimized burial depth–groundwater evaporation curve is more representative (Figure 3d).

### 3.2. Trend of Vegetation Coefficient with Depth of Groundwater Burial

The vegetation coefficient is a good reflection of the groundwater evaporation intensity characteristic in a given area. The results of fitting the data in Table 3 are shown in Figure 4, and the fit is good with an $R^2$ as high as 0.99 (Figure 4).

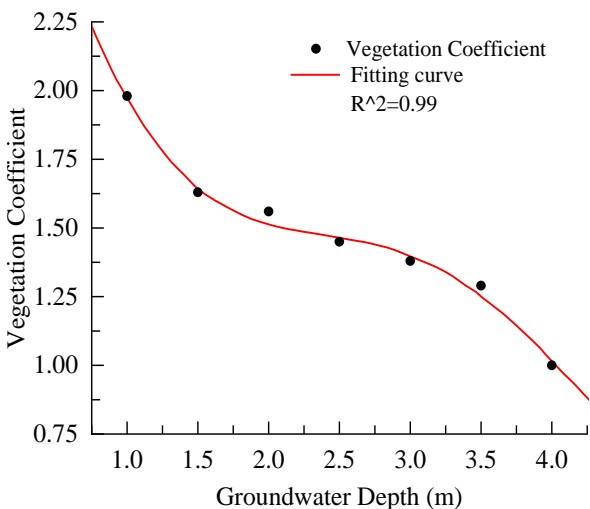

**Figure 4.** Trend of vegetation coefficient with depth of groundwater burial.

The *K* value of various vegetation was determined by the groundwater at different burial depths, and its value decreased with increasing groundwater burial depth (Figure 4). The influence of plants on soil moisture evaporation decreases with increasing depth to the water table. (1) When the depth of groundwater burial was greater than 2 m, the decrease rate of *K* became slower. This is due to the decrease in water due to the decrease in burial depth. (2) When the depth of groundwater burial increased between 2 and 3 m, the decrease rate remained stable because the soil moisture at that depth remained stable. (3) The rate increased rapidly when the depth of groundwater burial was greater than 3 m. This is due to the change in vegetation in that layer from trees and shrubs with a high transpiration rate to herbs with a low transpiration rate, resulting in a rapid decrease in *K*. This change was consistent with the trend of *K* (Figure 4). Therefore, we believe that soil water evaporation is the main factor affecting the ecological water demand of vegetation in arid areas when the burial depth is less than 3 m, and the change rate of *K* is consistent with the change rate of soil water content; when the water level drops below 3 m, transpiration will decrease significantly. When the burial depth is greater than 3 m, the effect of soil water evaporation on the ecological water demand of vegetation gradually decreases, while the transpiration of vegetation is enhanced by the decrease in the water table.

### 3.3. NVDA under Different Developmental Models

The optimum groundwater depth of burial for arbors in the Minqin area is 2–4.5 m, for shrubs 3–4 m, for herbal plants 4–4.5 m and for other forests 3.5–4.5 m. We used the middle value of each vegetation type as the calculated depth. Then, *K* was calculated by using Equation (6), *E* and *W* were obtained by using the method described in Section 2.3, and the results are shown in Table 6.

**Table 6.** Groundwater burial depth data for each vegetation type.

|  | Arbor | Shrubs | Herbals | Other Forests |
|---|---|---|---|---|
| Groundwater depth (m) | 2.0–4.5 | 3.0–4.0 | 3.0–4.5 | 3.5–4.5 |
| Calculation depth (m) | 3.3 | 3.5 | 3.7 | 4.0 |
| Vegetation coefficient ($K$) | 1.324 | 1.253 | 1.167 | 1.016 |
| Phreatic water evaporation ($E$) (mm) | 153.06 | 134.38 | 118.53 | 99.01 |
| Vegetation water requirement quota ($W$) (mm) | 202.65 | 168.38 | 138.33 | 100.59 |

Referring to previous studies [18], the values of each index were determined in the stabilization mode, restoration mode and optimization mode, taking into account the unique conditions of the arid area (Table 7).

**Table 7.** Values of indicators for different development models in 2017 and 2025.

| Indicators | Development Model in 2017 | | | Development Model in 2025 | | |
|---|---|---|---|---|---|---|
|  | Stability | Recovery | Optimization | Stability | Recovery | Optimization |
| $I_1$ | 34.33% | 39.33% | 44.33% | 53.60% | 58.60% | 63.60% |
| $I_2$ | 0.095 | 0.105 | 0.115 | 0.095 | 0.105 | 0.115 |
| $I_3$ | 0.040 | 0.050 | 0.060 | 0.040 | 0.050 | 0.060 |
| $I_4$ | 14% | 15% | 16% | 15% | 16% | 17% |
| $I_5$ | 62 | 57 | 52 | 59 | 54 | 49 |
| $I_6$ | 6 | 5 | 4 | 5 | 4 | 3 |
| $I_7$ | 4000 | 3800 | 3600 | 3800 | 3600 | 3400 |
| $I_8$ | 7.47% | 6.47% | 5.47% | 6.47% | 5.47% | 4.47% |
| $I_9$ | 3% | 5% | 7% | 5% | 7% | 9% |
| $I_{10}$ | 70% | 68% | 66% | 70% | 68% | 66% |
| $I_{11}$ | 3300 | 3300 | 3300 | 3300 | 3300 | 3300 |
| $I_{12}$ | 2850 | 2850 | 2850 | 2850 | 2850 | 2850 |
| $I_{13}$ | 2100 | 2100 | 2100 | 2100 | 2100 | 2100 |
| $I_{14}$ | 1950 | 1950 | 1950 | 1950 | 1950 | 1950 |
| $I_{15}$ | 0.9 | 0.91 | 0.92 | 0.91 | 0.92 | 0.93 |
| $I_{16}$ | 0.91 | 0.92 | 0.93 | 0.92 | 0.93 | 0.94 |
| $I_{17}$ | 0.43 | 0.44 | 0.45 | 0.64 | 0.65 | 0.66 |
| $I_{18}$ | 40% | 42% | 44% | 66% | 68% | 70% |

According to the actual situation in Minqin County, the values of each indicator under different development models (Table 7) and the NVDA values of the main vegetation types in 2017 and 2025 (Tables 8 and 9) were obtained. We see that WRCC varies with the development model. The stable model has the smallest carrying area for vegetation, and the optimized model has the largest. Vertically, the arbor forests have the smallest carrying area, and the shrub forests have the largest carrying area. The main reason is that as secondary water use increases and farm irrigation becomes more efficient, the amount of water used by primary industry and secondary industry and for domestic purposes continues to decrease, thus allowing more water resources to be used for ecological protection and the tertiary industry. Minqin is a typical arid area with deeper buried groundwater, which is more suitable for shrubs, herbs and other dwarf vegetation.

**Table 8.** The 2017 design carrying area of major vegetation types.

|  | Stability | Recovery | Optimization |
|---|---|---|---|
| Arbor forest area (hm$^2$) | 6204.22 | 7064.08 | 7976.10 |
| Shrub forest area (hm$^2$) | 391,451.98 | 445,703.98 | 503,247.95 |
| Herbaceous area (hm$^2$) | 240,138.00 | 273,419.13 | 308,719.74 |
| Other forest area (hm$^2$) | 24,134.20 | 27,479.00 | 31,026.76 |

**Table 9.** The 2025 design carrying area of major vegetation types.

| | Stability | Recovery | Optimization |
|---|---|---|---|
| Arbor forest area (hm$^2$) | 7026.50 | 7869.03 | 8717.09 |
| Shrub forest area (hm$^2$) | 443,333.45 | 496,492.24 | 549,999.91 |
| Herbaceous area (hm$^2$) | 271,964.92 | 304,575.42 | 337,399.95 |
| Other forest area (hm$^2$) | 27,332.85 | 30,610.25 | 33,909.16 |

*3.4. Water Ecological Status of Natural Vegetation under Different Developmental Patterns*

To make the evaluation of ecological water consumption in a given year more rational, the ecological footprint and ecological carrying capacity of each vegetation type were determined by using Equations (25) and (26). Then, the water ecological deficit (or surplus) and WREPI were obtained by using Equations (27) and (28), and the results are shown in Table 10.

**Table 10.** Water ecological deficit (surplus) and WREPI for each type of vegetation under each development model.

| Vegetation Type | Development Mode | WRCC | WEF | Water Ecological Deficit (Surplus) | WREPI |
|---|---|---|---|---|---|
| | Stability | 0.44 | | 0.08 | 0.81 |
| Arbor area | Recovery | 0.66 | 0.36 | 0.30 | 0.55 |
| | Optimization | 0.90 | | 0.54 | 0.40 |
| | Stability | 0.37 | | 0.06 | 0.84 |
| Shrub area | Recovery | 0.55 | 0.31 | 0.24 | 0.57 |
| | Optimization | 0.74 | | 0.43 | 0.42 |
| | Stability | 0.30 | | 0.09 | 0.70 |
| Herbage area | Recovery | 0.45 | 0.21 | 0.24 | 0.47 |
| | Optimization | 0.61 | | 0.40 | 0.34 |
| | Stability | 0.22 | | −0.01 | 1.05 |
| The other forest area | Recovery | 0.33 | 0.23 | 0.10 | 0.70 |
| | Optimization | 0.44 | | 0.21 | 0.52 |

With the optimization of the development model, the water resource carrying capacity of all types of vegetation has increased. Compared with the water carrying capacity of the stable development model for the whole county in 2025, vegetation other than arbors, shrubs and herbs is in a water ecological deficit. The design water supply of the stable model is not sufficient to support the vegetation growth, and the water allocation according to this model cannot meet the sustainable development demand of all vegetation in the year. If water resources are allocated according to the restoration model and the optimization model, arbors, shrubs, herbage plants and other forests will be in an ecological surplus, indicating that the WRCC of this development model is sufficient to support the area of natural vegetation in Minqin County in that year. This suggests that these two development models are relatively sustainable.

The WREPI values of each vegetation type under different development models in 2025 were calculated to scientifically evaluate the ecological security status of each vegetation type (Table 10). If water resources were allocated under the stable mode, the WREPI of other forests was 1.05, which was in a high ecological warning state. The WREPIs of forests and shrubs were 0.81 and 0.84, respectively, which were in a low ecological warning state. The WREPI of herbaceous plants was 0.70, which was in a low ecological security state. Under the restorative development model, all vegetation types were in an ecologically safe state, with the WREPI of herbaceous plants being 0.47 (intermediate ecological safety state), while the WREPIs of arbors, shrubs and other forest trees were all in an intermediate ecological safety state. In the optimization model, the WREPI of arbors, shrubs and herbaceous plants were all in the medium ecological safety state, except for the WREP of other forests, which was 0.52 and in the low ecological safety state.

## 4. Discussion

### 4.1. Feasibility of Predicting Natural Vegetation Design in Arid Areas Based on NVDA

This study adopts an NVDA-based research method to provide scientific information for the sustainable development of water resources and natural vegetation restoration in Minqin County. Minqin County, Gansu Province, is the green barrier of Wuwei City. Minqin Oasis blocks the encirclement of the Badain Jaran Desert and Tengger Desert, which has an important strategic position for the environmental construction of Wuwei, Hexi and even the whole northwest region. Therefore, the water resources in the Minqin area have been the focus of scholars and governments at home and abroad, most of which involve quantitative studies on ecological water demand [38]. The average annual ecological water demand of Minqin is $75.54 \times 10^5$ m$^3$, including $49.15 \times 10^4$ m$^3$ for arbors, $22.50 \times 10^5$ m$^3$ for shrubs, $24.49 \times 10^5$ m$^3$ for other forests and $18.65 \times 10^5$ m$^3$ for herbaceous plants [39]. The average annual ecological water shortage is $21.54 \times 10^8$ m$^3$, of which $17.29 \times 10^6$ m$^3$ for arbors, $75.72 \times 10^6$ m$^3$ for shrubs, $75.68 \times 10^6$ m$^3$ for other forests and $46.68 \times 10^6$ m$^3$ for herbaceous plants [39]. These studies can provide valuable data to support the efficient allocation of water resources in Minqin County. However, in arid regions where water resources are scarce, it is relatively one-sided to allocate limited water resources according to the amount of water [40]. This study takes the principle that the type of vegetation planted is compatible with the type of land, combined with the concept that water resources should be allocated according to the ecological water for natural vegetation first and then for production and economic water, forming a "water-fixing forest" at the junction of two deserts in Minqin County, which is of great significance to the ecological recovery of arid areas. In addition, although the water ecological footprint evaluation method based on WRCC can qualitatively analyze the rationality of water resource allocation in arid areas to a certain extent, there are certain errors in the process of model operation [41,42]. Therefore, in the future, we can combine the water ecological footprint evaluation model based on WRCC with the satellite remote sensing technology of GIS to realize the information integration of water resource usage, vegetation type and coverage area and land use change, so as to provide ecological sustainable development solutions with higher prediction accuracy, more comprehensive information and more reasonable management methods for the design of natural vegetation areas in arid areas based on WRCC [43,44].

### 4.2. Strategies for Meeting the Sustainable Supply of Regional Water Resource Carrying Capacity

The area of various types of natural vegetation in the Minqin area based on WRCC prediction will be further expanded, and the water resource allocation under the restoration mode and optimization mode will alleviate the local ecological sustainability problem to a certain extent, which is also in line with the actual water use law [17]. First of all, in China, agriculture is the most water-consuming industry, and its production requires about 61% of the total water resources. However, with the further acceleration of agricultural modernization and the application and popularization of various water-saving irrigation technologies, the effective utilization coefficient of agricultural irrigation water has increased by 0.052 in the past decade. "The Fourteenth Five-Year Plan of Minqin County Development Plan" mentions that the effective utilization coefficient of irrigation water in Minqin County will reach 0.63 in 2025, which will provide more water resources for natural vegetation recovery and ecological restoration in the Shiyang River Basin and Minqin County [45]. Secondly, China's current water rights system adjusts the overall water market allocation scheme, so that the allocation of water resources will be more reasonable. The groundwater level continues to rise, and the ecological water consumption of natural vegetation is further guaranteed, but also the protection and development rights for water users who are in a disadvantaged position in terms of water use are provided, which improves the utilization rate of water resources in the region and makes the development of various industries more balanced and smooth [46]. Finally, the implementation of ecological protection policies by government departments will provide a strong impetus for the growth of natural vegeta-

tion areas and ecological security. For example, in October 2021, "*The Outline of the Plan for Ecological Protection and High-Quality Development of the Yellow River Basin*" was issued and implemented. A series of major strategic policy documents on watershed ecological protection, such as "*The Yellow River Protection Law*", which came into effect in April 2023, are the cornerstones of sustainable development and will ultimately create a virtuous water cycle.

*4.3. Reliability of the Results of Natural Vegetation Area Calculation Based on NDVI*

From the calculation results, it can be seen that the water resource allocation scheme in the Minqin area in 2017 and 2025 can achieve the restoration of natural vegetation and positive ecological evolution. With the optimization of the development pattern, the water resource carrying area of vegetation increases accordingly, but the water ecological pressure on shrubs, arbors and other forests remains strong, and the current water resources cannot further increase the abundance and number of their species to ensure the benign evolution of vegetation. Therefore, it is crucial to develop the following measures to solve the temporal contradiction between the amount of water resources and vegetation area: (1) when water resources are sufficient, increase the water consumption of shrubs, arbors and other forests to relieve water ecological pressure; (2) in the case of water shortage, transfer the right amount of water resources from herbaceous species to shrubs, arbors and other forests to achieve a balance between water supply and demand; and (3) promote water conservation techniques and concepts, which will include more efficient irrigation techniques and viable mulching techniques to reduce inefficient water losses in the Minqin area and domestic production. Through the above, we will balance the relationship between water use for living, production and ecology, so as to realize the positive and smooth development of the "three livelihoods" and to make more efficient and ecological use of limited water resources as far as possible [47]. Analysis of future water resources is often difficult due to stochastic factors. In addition, the predicted results are uncertain due to the uncertainty of these resources. Using design water resources with a certain guarantee rate can reduce this uncertainty while simplifying the calculation of natural vegetation carrying area [48]. However, when estimating groundwater evaporation, the accuracy of the calculation results depends largely on the appropriateness of the chosen equations and the accuracy of the vegetation coefficients. Factors such as soil texture, groundwater burial depth, soil water evaporation and vegetation type can all affect the calculation results [49]. To eliminate possible errors between estimated and actual values, it is necessary to observe and determine groundwater evaporation parameters for different groundwater burial depths in the future.

**5. Conclusions**

In this study, based on the systematic analysis of the existing studies on water resource carrying capacity evaluation in arid areas, the concept and calculation method of natural vegetation water resource design carrying area in arid areas were proposed, as well as the evaluation of the reasonableness of the calculation results based on water ecological footprint, water ecological deficit and water resource pressure index. It also predicts the natural vegetation area that can be carried by water resources in the arid area of Minqin County in northwest China in 2025. The area of natural vegetation (arbors, shrubs, herbs and other forests) in the county under each development model has a relatively large increase in 2025 relative to 2017, according to the characteristics of local vegetation distribution. Among them, the degree of water resource utilization in Minqin County under the stable development model is about to exceed the critical value, and although the water resource ecological pressure index of arbors, shrubs and herbs is within the safe range, the water resource consumption of other forests is greater than the maximum exploitable value and is in an unsafe state. The water resource utilization of Minqin under the restoration mode and the optimization mode is in a safe state, which is in line with the natural water use law and can be used as the preferable choice for the ecological sustainable

development plan in the study area. In addition, the quantity and abundance of natural vegetation have a significant impact on the ecological benign evolution of the arid area and also contribute to the further development of agriculture to improve the livelihoods of local populations. When the shortage of available water resources contradicts vegetation water use, the balance of water supply and demand between vegetation types can be properly coordinated, thus forming a benign water cycle and ecological security guarantee.

**Author Contributions:** Data curation, T.J., C.Z. and F.L.; formal analysis, T.J., J.Y. and X.C.; funding acquisition, H.Z.; writing—original draft, T.J. and J.Y.; supervision, H.Z. and S.Y.; writing—review and editing, J.Y., H.Z. and T.J. All authors have read and agreed to the published version of the manuscript.

**Funding:** This work was supported by the National Natural Science Foundation of China (No. 51669001, 52269008), the Open Project of Liaocheng University Landscape Architecture Discipline (No. 31946221236), the Industrial Support Plan Project of Gansu Provincial Department of Education (No. 2022CYZC-51) and the Key Research and Planning Projects of Gansu Province (No. 18YF1NA073).

**Informed Consent Statement:** Informed consent was obtained from all subjects involved in the study.

**Data Availability Statement:** The datasets used and/or analyzed during the current study are available from the corresponding author upon reasonable request and the approval of the data owner.

**Acknowledgments:** We thank everyone who helped during the manuscript writing. We especially thank Tianliang Jiang for his preliminary important work. We also thank the reviewers for their useful comments and suggestions.

**Conflicts of Interest:** The authors declare no conflict of interest.

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
