# Peer review of "Natural Vegetation Area Design in an Arid Region Based on Water Resource Carrying Capacity—Taking Minqin County as an Example"

_water, doi:10.3390/w15183238_

Round 1

Reviewer 1 Report

Comments and Suggestions for Authors

General Comments
the authors designed the article to study the “Natural Vegetation Area Design in an Arid Region Based on Water resource Carrying Capacity—Taking Minqin County as an example”.

 The article has a lot of discrepancies and needs improvement before further consideration.

Specific Comments:

The article abstract needs to be improved and add the quantify results according to methodology.

The introduction section needs to be carefully written and add more literature studies. Add more data and novelty statement in the last paragraph of the introduction section.

Provide proper citations in the table-2 for each equation/formula.

From equation 7 to equation 24, please elaborate each equation. Maybe after every equation or at the end of the all the equations. There are many parameters that are missing and not elaborated properly.

In table 4, please provide proper caption of each column.

In table 7, can you provide the accuracy level of each Indicators, to showcase the difference in each Indicator level about predicting the vegetation by each developed model. Therefore, maybe some models are predicting some good results.

Please strengthen your discussion by citing more related studies.

Kindly revise the article conclusion.

Comments on the Quality of English Language

Minor editing of English language required.

Author Response

Response to Reviewer 1

Dear Reviewer 1,

Thanks very much for taking your time to review this manuscript. We really appreciate all your comments and suggestions! Please find our itemized responses below and our revisions in the re-submitted files.

Thanks again!

*************************

Reviewer 1:

Questions: (1) The article abstract needs to be improved and add the quantify results according to methodology.

Answer: Thank you for underlining this deficiency. Authors have revised in the manuscript (Line 12-36).

Questions: (2) The introduction section needs to be carefully written and add more literature studies. Add more data and novelty statement in the last paragraph of the introduction section.

Questions: (3) Provide proper citations in the table-2 for each equation/formula.

Answer: Thank you for underlining this deficiency. Authors have revised in the manuscript (table-2).

Questions: (4) From equation 7 to equation 24, please elaborate each equation. Maybe after every equation or at the end of the all the equations. There are many parameters that are missing and not elaborated properly.

Answer: Authors have modified your suggestion by adding a note at the end of the equation, while Figure 2 includes a partial additional description of equation 7 to equation 24.

Questions: (5) In table 4, please provide proper caption of each column.

Answer: Authors have made changes to the suggestions you made. The header column in Table 4 has been redefined to meet the type description of the data in the table.

Questions: (6) In table 7, can you provide the accuracy level of each Indicators, to showcase the difference in each Indicator level about predicting the vegetation by each developed model. Therefore, maybe some models are predicting some good results.

Answer: Thank you very much for this point, but the authors have provided explanatory notes in lines 158-173 of the manuscript, which hopefully make clear the differences in the levels of indicators used by the models to predict vegetation.

Questions: (7) Please strengthen your discussion by citing more related studies.

Answer: Thank you for underlining this deficiency. The authors have added literature citations to the discussion section of the manuscript to enhance the scientific nature of the discussion (Line 420-507).

Questions: (8) Kindly revise the article conclusion.

Answer: Thank you very much for your constructive comments. The authors have referred to the work done in this manuscript, the findings and important conclusions in the concluding part of the manuscript, and finally, they have made recommendations for balancing the consumption of water in the context of water resources carrying capacity in all its dimensions, so as to ensure ecological security and sustainable development of mankind.

Appended to this letter is our point-by-point response to the comments raised by the reviewers.

We would like also to thank you for allowing us to resubmit a revised copy of the manuscript.

We hope that the revised manuscript could be accepted for publication in the Water.

Sincerely,

Authors with Manuscript ID: water-2560320

Reviewer 2 Report

Comments and Suggestions for Authors

Comments on the Quality of English Language

Some of the terminology might be modified to better relate to terminology used in most western journal papers.

Author Response

Response to Reviewer 2

Dear Reviewer 2,

Thanks very much for taking your time to review this manuscript. We really appreciate all your comments and suggestions! Please find our itemized responses below and our revisions in the re-submitted files.

Thanks again!

*************************

Reviewer 2:

Questions: (1) Ln 96 – not clear what is meant by sanding area – is that the active area of sand deposition resulting from movement resulting from an area as it dries even more.

Answer: Thank you for underlining this deficiency. The authors have revised the relevant sentence in line 101.

Questions: (2) Ln 103 – is the 4.5 m the depth of the saturated zone below the surface of the desert or is it the depth to the groundwater table?

Answer: The depth of the saturated zone below the desert surface is 4.5 meters.

Questions: (3) Ln 112 – by design of the natural vegetation area do you mean the actual design of species identification and placement within the area of interest?

Answer: Through the calculation of water resources carrying capacity on the area of all kinds of vegetation in the sky vegetation area, we can understand the carrying area and development direction of all kinds of natural vegetation in Minqin County, and at the same time, we can provide theoretical guidance for a more reasonable distribution of water resources in Minqin County in the future.

Questions: (4) Ln 116 – for readers not familiar with Chinese currency – it would be helpful to spell out Chinese Yuan.

Answer: Thank you for the suggestion. The authors have revised in the manuscript (Line 121-139).

Questions: (5) Ln 122 – total available water resources and available water supply are the same values so the sentence could be consolidated to reflect that.

Answer: Thank you for underlining this deficiency. Authors have revised in the manuscript (Line 127-129).

Questions: (6) Ln 134 – “the maximum carrying area of water resources for natural vegetation...”is not clear to me. What is meant by water carrying area – the available water in a specific volume of area?

Answer: "the maximum carrying area of water resources for natural vegetation..." means how large an area of natural vegetation can be supported by the available water resources in the area.

Questions: (7) Ln136 – the explanation of “design flood” is not clear to me – is it the size of flood that the projected human development of the area could withstand but it would require dams and flood walls, etc. which are suggested by the phrase “require reinforcement.” What is meant by “general flood” is it just any flood that might occur while the design one is specific to a landscape and its unique features?

Answer: Thank you for underlining this deficiency. The "design flood" is only an example of the authors' approach, which is similar to that of the "design water supply", and the authors have refined the content of what is covered in the manuscript (Line 143).

Questions: (8) Ln 140 – NVDA refers to the impact of the unique features on a given landscape and their impact on the size of flood it can handle.

Answer: The NVDA is constantly changing with the natural landscape features and is influenced by the given unique landscape features of the study area.

Questions: (9) Ln 149 – the water supply guarantee rate is the volume or quantity of water needed to support a residential or industrial area? I am not a modeler so am not clear on why this guarantee rate is a percent – does the 95% suggest that a residential area needs to be assured that it will always have 95% of the design water needs available?

Answer:

1) Thank you very much for this question. In this context, water supply guarantee rate refers to the probability that a water resource quantity may be available for a certain period of time in the future for the designed water supply, and also expresses the likelihood that a particular water resource quantity will be met.

2) The 95% refers to the probability that the design water supply is likely to have water resources within a certain period of time in the future.

Questions: (10) Ln 167 – If I understand this correctly the optimization mode is suggesting that the plant community that is reestablished or maintained on a site is designed to fit with the water that is available after the uses of that water for maintaining the human needs of the area have been established?

Answer: Your understanding of the optimization model is consistent with what the authors have expressed. The optimization mode that the type and quantity of regional vegetation can reach the maximum carrying capacity. At the same time, the ecological and environmental functions can give full play to their potential.

Questions: (11) Ln 180 – Figure 2 is very helpful in providing a summary of all the variables that are using water in a specific area to help determine whether the available water in the area can ultimately meet the needs of the area.

Answer: Thank you very much for this suggestion. Figure 2 expresses in detail the Minqin County water use system evaluation indicators in terms of socio-economic, ecological and water use.

Questions: (12) Ln 184 – soil moisture evaporation tends to decrease even without a vegetation cover as water in pores near the surface are dried and water must be evaporated from pores that extent further down in the soil.

Answer: Vegetation cover was only one of the factors affecting evaporation of surface soil moisture. Soil moisture evaporation tends to decrease even without a vegetation cover as water in pores near the surface are dried and water must be evaporated from pores that extent further down in the soil.

Questions: (13) Ln 185 – not sure what is meant by “groundwater burial depths” is that plant rooting depths? And depending on the kind of vegetation and soil cover evaporation from the soil surface can vary greatly for example, if there is a dense forest floor cover of decomposing leaf material or even depending on the type of vegetation cover – large open grown trees versus very dense grasses and forbs in a healthy native prairie community.

Answer: Vegetation cover affects the rate and intensity of surface water evaporation, and the meaning of the vegetation storage quota expressed by the authors of the manuscript was calculated by the product of the evaporation intensity of groundwater and the vegetation coefficient, which in this case was not directly related to the rooting depth of plants.

Questions: (14) Ln 191 – isn’t groundwater evaporation intensity is controlled by porosity of the soil, depth to water in the soil pores and humidity of the air?

Answer: Thank you very much for this question. The intensity of groundwater evaporation is controlled by the porosity of the soil, the depth of water in the soil pores and the humidity of the air. What the authors were expressing here was that evapotranspiration can be calculated in two ways, and both methods incorporate the soil pores, the depth of who was in the soil pores, and the humidity of the air.

Questions: (15) Ln 223 – I apologize that I cannot comment fully on the presentations of equations between Ln 191 and 223 and assume that it all makes sense.

Answer: The equations between line191-223 were used to determine the intensity of groundwater evaporation through a process of optimization and fitting, which involves the validation and selection of classical equations, and fitting, which includes steps such as correlation and difference tests with classical equations.

Questions: (16) Table 3 – units of measure are missing – not sure what goes into the vegetation coefficient – what does it represent in terms of the vegetation, rooting design, depths and extents for example?

Answer:

1) Thank you for underlining this deficiency. Authors have revised in the manuscript (Table 3).

2) The vegetation coefficient (?) reflects the influence of vegetation on groundwater evaporation in a certain area, and is obtained by dividing the groundwater evaporation in the vegetated area by that in the non-vegetated area.

Questions: (17) Ln 263 &264 – it is not clear to me what is meant by land area needed to absorb water pollution? What kind of pollution and how is that polluted water applied to the land – does this refer to surface runoff from urban areas, runoff that may end up running over the surface of the land – does surface runoff not usually concentrate into depressions and end up in surface waters even in arid areas where ephemeral streams may only be active for short periods after precipitation events? I assume what is meant by sustainable water consumption is the biologically productive area from which water can be extracted to meet population and associated supporting systems demands?

Answer: The water ecological footprint is an indicator for evaluating the status of non-point source pollution and is defined as the biologically productive land area necessary to sustain water consumption and absorb water pollution for a given population and economic size, which in this case includes the area of land required by nature for the process of wastewater uptake and self-purification. At the same time, sewage carries pollutants that affect the normal use of the soil, which requires the long-term action of soil chemicals and micro-organisms to achieve soil ecological security.

Questions: (18) Ln 293 – I am still confused – in these calculation results is groundwater evaporation intensity include water loss from transpiration of vegetation covering the volume of land being assessed?

Answer: We apologize for the lack of clarity in the manuscript regarding the meaning of evapotranspiration intensity. The evapotranspiration intensity described in this study includes evapotranspiration from natural vegetation areas, and water resources applied to natural vegetation areas (arid zones with deep groundwater) will be consumed in both ways.

Questions: (19) Ln 322 – By groundwater throughout the paper, I have been assuming it is water below the water table so it would seem obvious that the influence on soil water evaporation by plants would decrease as water table depths increased. The statement is made that K changes rapidly when dominant vegetation changes from arbors and shrubs to herbaceous vegetation. However, in native tall-grass prairie systems roots can be found down to 4 meters allowing high rates of ET from a site. Trees growing on drier sites tend to have tighter control of stomates which can reduce demand for water for transpiration.

Answer: Thank you for underlining this deficiency. Authors have revised in the manuscript (Line 349-350). The influence of plants on soil moisture evaporation decreases with increasing depth to the water table.

Questions: (20) Ln 331 – you are experts in dry arid regions but it would seem to me that as the water table drops below 3 m transpiration would be decreased significantly (Ln) – I can see that ecological water demand would gradually decrease because of decreasing plant density and better control of stomates by more arid site plant species.

Answer: Thank you for underlining this deficiency. Authors have revised in the manuscript (Line 360-361). When the water level drops below 3 m, transpiration will decrease significantly.

Questions: (21) Ln 347 – what are the values for 2025 based on?

Answer: The values for 2025 based on the "Notice of the People's Government of Minqin County on the Issuance of the Fourteenth Five-Year Plan of National Economic and Social Development of Minqin County and the Outline of Visionary Goals for 2035", by 2025, the total available water resources and the available water supply of the county will be 4.13×108 m3.

Questions: (22) Ln 357 – I still have a problem with the statement that the deeper groundwater is more suitable for shrubs, herbs and other dwarf vegetation. I can agree with the statement if it is mentioned that these suggested species are very different than the shrubs, herbs and dwarf vegetation that is commonly found in forest ecosystems.

Answer: The natural vegetation mentioned in the manuscript includes shrubs, herbs, and other dwarf plants, which encompass the main natural vegetation in Minqin County, which is located in the middle of China's two major deserts and has a relatively low forest distribution, and thus the species mentioned in the manuscript are quite different from the shrubs, herbs, and dwarf vegetation commonly found in forest ecosystems.

Questions: (23) Ln 390 – this is an interesting article addressing a real challenging issue for many dry regions of the world. While the modelling is probably excellent, as I mentioned earlier, I am not a modeller, one of my concerns is that only very general plant forms are used, those being forests, shrubs, herbs and dwarf vegetation but within each of these broad classes of plants are species that vary dramatically between different water available landscapes. Cactus as large as trees and as small as small herbaceous plants can exist in arid climates and deserts because of evolved modifications of their anatomy. So I am a bit concerned with the generalizations of ranking these very broad plant groups by water use needs – it seems somewhat over simplified.

Answer: We very much agree with and appreciate your views and the comments made. Although the study in the manuscript is relatively simplified, in Minqin County, the natural vegetation consists mainly of arbors, shrubs, herbs, and other dwarf plants, with a relatively small distribution of other arid desert-growing plants such as cacti. A generalization of these arid desert-grown plant taxa sorted by water demand will be one of our next research priorities.

Questions: (24) Ln 392 – Discussion – The message here seems to be that more careful use of a limited water supply will provide more “ecological water” to support native vegetation that will help to stabilize a landscape that is very fragile and susceptible to massive wind erosion. The argument of the authors seems to be that more careful use of water for human needs will leave more water available for native vegetation that will help to stabilize the soil and hold it in place, but that this will only happen if very limited water is protected. From other reading It sounds as though forest cover in Minquin county has increased from about 3% in 1950 to 18% in 2021 and that seems partly due to more careful use of water for large areas devoted for irrigated agriculture to the point that water consumption for some agricultural products has been reduced by up to 40% according to a China Daily article in 2021. The material carefully presented in this paper provides the scientific background to improve water availability even more with the selection of water efficient native vegetation. It seems quite optimistic that the effective utilization coefficient of irrigation water in the county can go frm 0.052 in the past decade to 0.63 in the next two years – that seems very optimistic.

Answer: The ecological location of Minqin County is extremely important, and "Minqin must not be allowed to become the second Lop Nor" has been widely recognized. Therefore, we should be more careful with our limited water supply, and use more "ecological water" to support native vegetation, which will help to stabilize a very fragile and sandy landscape that is susceptible to large-scale wind erosion.

Questions: (25) Ln 460 – it is interesting that the authors now suggest as their first measure to solve the temporal contradiction between the amount of water resources and vegetation area that they recommend increasing the amount of water used by woody plants. They do also recommend further improvement of water conservation techniques which I would assume would include more efficient irrigation techniques and possibly mulching.

Answer: Thank you very much for your suggestions for solving the water shortage problem and water allocation issues in Minqin County, and the authors are similar views with you, as well as revising and improving the relevant parts of the manuscript discussion (Line 421-512).

Questions: (26) Ln 477 – the recommendation to further increase the amount of natural vegetation in the county should not only help reduce the potential of destructive sand storms but also help to further and develop an agricultural sector to improve the livelihoods of the local population.

Answer: Thank you very much for your analysis of the merits of natural vegetation in Minqin County when its abundance increases. The authors' views are similar to yours: further increasing the abundance of natural vegetation in the county will not only help to reduce potentially damaging dust storms and sandstorms, but will also help to further develop the agricultural sector and improve the livelihoods of local people, as well as revising and improving the relevant parts of the manuscript conclusions (Line 513-536).

Appended to this letter is our point-by-point response to the comments raised by the reviewers.

We would like also to thank you for allowing us to resubmit a revised copy of the manuscript.

We hope that the revised manuscript could be accepted for publication in the Water.

Sincerely,

Authors with Manuscript ID: water-2560320

Round 2

Reviewer 1 Report

Comments and Suggestions for Authors

Questions: (2) The introduction section needs to be carefully written and add more literature studies. Add more data and novelty statement in the last paragraph of the introduction section. Please address this comment.

Comments on the Quality of English Language

Minor English corrections and Grammer check before final publication.

Author Response

Response to Reviewer 1

Dear Reviewer 1,

Thanks very much for taking your time to review this manuscript. We really appreciate all your comments and suggestions! Please find our itemized responses below and our revisions in the re-submitted files.

Thanks again!

*************************

Reviewer 1:

Questions: (1) The article abstract needs to be improved and add the quantify results according to methodology.

Answer: Thank you for underlining this deficiency. Authors have revised in the manuscript (Line 20-44).

Questions: (2) The introduction section needs to be carefully written and add more literature studies. Add more data and novelty statement in the last paragraph of the introduction section.

Questions: (3) Provide proper citations in the table-2 for each equation/formula.

Answer: Thank you for underlining this deficiency. Authors have revised in the manuscript (table-2).

Questions: (4) From equation 7 to equation 24, please elaborate each equation. Maybe after every equation or at the end of the all the equations. There are many parameters that are missing and not elaborated properly.

Answer: Authors have modified your suggestion by adding a note at the end of the equation, while Figure 2 includes a partial additional description of equation 7 to equation 24.

Questions: (5) In table 4, please provide proper caption of each column.

Answer: Authors have made changes to the suggestions you made. The header column in Table 4 has been redefined to meet the type description of the data in the table.

Questions: (6) In table 7, can you provide the accuracy level of each Indicators, to showcase the difference in each Indicator level about predicting the vegetation by each developed model. Therefore, maybe some models are predicting some good results.

Answer: Thank you very much for this point, but the authors have provided explanatory notes in lines 179-197 of the manuscript, which hopefully make clear the differences in the levels of indicators used by the models to predict vegetation.

Questions: (7) Please strengthen your discussion by citing more related studies.

Answer: Thank you for underlining this deficiency. The authors have added literature citations to the discussion section of the manuscript to enhance the scientific nature of the discussion (Line 437-527).

Questions: (8) Kindly revise the article conclusion.

Answer: Thank you very much for your constructive comments. The authors have referred to the work done in this manuscript, the findings and important conclusions in the concluding part of the manuscript, and finally, they have made recommendations for balancing the consumption of water in the context of water resources carrying capacity in all its dimensions, so as to ensure ecological security and sustainable development of mankind.

Appended to this letter is our point-by-point response to the comments raised by the reviewers.

We would like also to thank you for allowing us to resubmit a revised copy of the manuscript.

We hope that the revised manuscript could be accepted for publication in the Water.

Sincerely,

Authors with Manuscript ID: water-2560320